# Prevalence and Determinants of Fatigue after COVID-19 in Non-Hospitalized Subjects: A Population-Based Study

**DOI:** 10.3390/ijerph18042030

**Published:** 2021-02-19

**Authors:** Knut Stavem, Waleed Ghanima, Magnus K. Olsen, Hanne M. Gilboe, Gunnar Einvik

**Affiliations:** 1Institute of Clinical Medicine, University of Oslo, 0450 Oslo, Norway; waleed.ghanima@so-hf.no (W.G.); gunnar.einvik@medisin.uio.no (G.E.); 2Health Services Research Unit, Akershus University Hospital, 1478 Lørenskog, Norway; 3Department of Pulmonary Medicine, Division of Medicine, Akershus University Hospital, 1478 Lørenskog, Norway; 4Department of Medicine, Østfold Hospital Trust, 1714 Grålum, Norway; 5Department of Research, Østfold Hospital Trust, 1714 Grålum, Norway; magnus.kringstad.olsen@so-hf.no; 6Fürst Medical Laboratory, 1051 Oslo, Norway; hmgilboe@furst.no

**Keywords:** COVID-19, fatigue, post-infectious fatigue, prevalence, CFQ-11

## Abstract

This study assessed the prevalence and determinants of fatigue in a population-based cohort of non-hospitalized subjects 1.5–6 months after COVID-19. It was a mixed postal/web survey of all non-hospitalized patients ≥18 years with a positive PCR for SARS-CoV-2 until 1 June 2020 in a geographically defined area. In total, 938 subjects received a questionnaire including the Chalder fatigue scale (CFQ-11) and the energy/fatigue scale of the RAND-36 questionnaire. We estimated z scores for comparison with general population norms. Determinants were analyzed using multivariable logistic and linear regression analysis. In total, 458 subjects (49%) responded to the survey at median 117.5 days after COVID-19 onset, and 46% reported fatigue. The mean z scores of the CFQ-11 total was 0.70 (95% CI 0.58 to 0.82), CFQ-11 physical 0.66 (0.55 to 0.78), CFQ-11 mental 0.47 (0.35 to 0.59) and RAND-36 energy/fatigue −0.20 (−0.31 to −0.1); all CFQ-11 scores differed from those of the norm population (*p* < 0.001). Female sex, single/divorced/widowed, short time since symptom debut, high symptom load, and confusion during acute COVID-19 were associated with higher multivariable odds of fatigue. In conclusion, the burden of post-viral fatigue following COVID-19 was high, and higher than in a general norm population. Symptoms of fatigue were most prevalent among women, those having a high symptom load, or confusion during the acute phase.

## 1. Introduction

Most patients with COVID-19 experience none or only minor upper respiratory symptoms, but respiratory symptoms, such as cough or dyspnea, and non-respiratory symptoms, such as fatigue, tiredness, sleepiness, and headache, are common during the acute phase of COVID-19 [1,2]. In several other infectious diseases such as Epstein–Barr virus, Q fever, Ross River virus, giardiasis, or in community-acquired pneumonia, such symptoms may persist several months after an infection [3,4]. During follow-up in survivors of other coronaviruses, such as severe acute respiratory syndrome (SARS), 64% reported fatigue at 3 months, 54% at 6 months, and 60% at 12 months [5,6]. Following Middle East respiratory syndrome (MERS), 48% had clinically relevant fatigue after 12 months [7].

There is considerable concern about long-term sequelae following COVID-19 and that the disease will trigger post-viral fatigue syndromes [8,9,10,11]. The prevalence of fatigue following hospitalization for COVID-19 ranges from 52 to 70% at 1–3 months after hospital discharge [12,13,14]. Although non-hospitalized patients represent a larger patient group, the majority of emerging data concerns hospitalized patients.

In a convenience sample of non-hospitalized subjects, 50–75% reported symptoms of fatigue during COVID-19 [15], but little information is available with longer follow-up. In a recent population-based study, 24% reported persistence of fatigue at follow-up by telephone [16]. The above studies used single items for assessing fatigue. There is little information available using more comprehensive multi-item questionnaires designed for assessment of fatigue, with the exception of a recent study comprising a combination of admitted patients and outpatients [17], which reported that, overall, 52% had fatigue as assessed with the Chalder fatigue scale (CFQ-11) [18]. Using such instruments also enables comparison with general population norms, which eases interpretation of fatigue scores.

The objective of this study was to determine the prevalence of fatigue using the CFQ-11 in a large population-based cohort of non-hospitalized subjects in Norway, on average 4 months after COVID-19 infection, and to compare with general population norms, as well as determine risk factors for persistent fatigue.

## 2. Materials and Methods

### 2.1. Study Design and Population

This was a cross-sectional mixed-mode survey of a geographical cohort in the catchment areas of two Norwegian hospitals, Akershus University Hospital (Ahus) and Østfold Hospital (ØH), representing about 900,000 inhabitants in 2020, or 17% of the population of Norway.

Up to 1 June 2020, we identified 1029 polymerase chain reaction (PCR) SARS-CoV-2-positive subjects ≥18 years from the microbiology laboratories of Ahus, ØH, and the largest private microbiology laboratory in the region, Fürst Medical Laboratory. After the exclusion of 91 subjects (Figure 1), 938 were eligible for inclusion in the survey [19].

### 2.2. Mixed-Mode Survey and Questionnaire

The survey started as a postal survey at the end of June, 2020. The subjects were asked to sign a consent form on-line (web-push) by using their personal identification number through a national electronic identification system (ID-porten), and then were forwarded to an online web-questionnaire. Alternatively, they could sign the consent form and respond to a paper questionnaire with identical items, which were returned in postage-prepaid envelopes. After about 5 weeks, non-respondents received a postal reminder [19].

The questionnaire contained background information on demographics, education, previous physician-diagnosed disease, recent influenza immunization, date of COVID-19 symptom onset, symptoms during acute COVID-19, current symptoms, height, weight, and standardized questionnaires on health status and fatigue.

### 2.3. Assessment of Fatigue

Fatigue was assessed using the Chalder fatigue scale (CFQ-11) [18]. It contains 11 items on an ordinal 0 to 3 scale, which are summed to a total score (range 0–33; 33 denotes maximal symptoms) and two subscales: physical fatigue (seven items, range 0–21) and mental fatigue (four items, range 0–12). The CFQ-11 also has an alternative scoring algorithm, bimodal scoring, where each item response is dichotomized; 0 (0 to 1) or 1 (2 to 3) and summed to a 0–11 scale. Conventionally, fatigue case-status (fatigued vs. non-fatigued) is defined using this scale with a cut-off at <4 vs. ≥4 [18,20].

We assessed health status using the RAND-36 questionnaire [21]. In this analysis, we used the energy/fatigue scale of this questionnaire as a marker of fatigue [22]. It consists of four items, and is scored on a 0–100 scale, where 100 denotes maximal health.

### 2.4. Assessment of Comorbidity and COVID-19 Symptoms

Comorbidity was recorded using a checklist of 21 physician-diagnosed problems, 18 of which constituted a self-reported version of the Charlson comorbidity index [23,24] and some additional items of relevance in COVID-19. We categorized this comorbidity index as 0, 1, and ≥2 comorbidity. In addition, we used history of depression separately, as this variable is associated with fatigue [17,25].

As measures of the severity of the acute phase of COVID-19, we used: (1) a symptom score based on 23 self-reported symptoms, not including fatigue, i.e., the total number of self-reported symptoms categorized in tertiles (0–5, 6–9, and 10–23); (2) presence of dyspnea; and (3) presence of confusion. 

### 2.5. Statistical Analysis

Participants’ characteristics are presented as mean (SD), median (interquartile range or range) or number (%), as appropriate. Groups were compared using the *t*-test for continuous, or chi-squared test for categorical, variables.

The prevalence of fatigue was assessed using score ≥4 from bimodal scoring of the CFQ-11 [18] as cut-off. The crude prevalence of fatigue was compared between men and women using the chi-squared test.

In addition, we presented total fatigue score (0–33 range) and two dimension scores, as well as RAND-36 energy/fatigue scores as continuous scores. We calculated z scores for each participant on these scales, providing the fatigue level for each participant in comparison to mean and SD in the matching age- and sex-specific stratum in normative data from Norwegian general populations [20,26]. The z score represents the difference from the mean of the norm population, reported in number of SDs.

Determinants of fatigue (dichotomized CFQ-11 bimodal score, <4 vs. ≥4) were analyzed using multivariable logistic regression. Determinants of fatigue scores (CFQ-11 total, Likert scoring) and the Energy/fatigue scale of the RAND-36 were analyzed using multiple linear regression analysis.

We included the following independent variables in the models, based on the literature and what we thought might be important: age/10, sex, education (three levels), marital status (married/cohabiting vs. single/widowed), body mass index (BMI) (kg/m^2^), number of self-reported comorbidities, a modification of the Charlson index (0, 1, ≥2), smoking status (current/previous vs. never smoker), history of depression (yes vs. no), number of symptoms during acute COVID-19 (0–5, 6–9, 10–23), and dyspnea (yes vs. no) or confusion (yes vs. no) during COVID-19, as well as time since symptom onset during COVID-19 (tertiles; 41–109, 110–126, 127–182 days). All variables were entered into the multivariable models without any statistical variable selection procedure. As there were few missing values, we used complete case analysis.

We used Stata (version 16.1, Stata Corporation, College Station, TX, USA) for all statistical analyses, and chose a 5% a significance level.

## 3. Results

### 3.1. Study Population

In total, 458 subjects (49% of the gross sample, 51% of the net sample) completed the questionnaire (Figure 1). The respondents were older (mean age on 1 June 2020 49.5 (SD 15.3) vs. 43.9 (17.3) years, *p* < 0.001) and comprised a larger proportion of women (256 (56%) vs. 219 (46%), *p* = 0.005), than non-respondents in the gross sample.

Respondents’ characteristics are shown in Table 1. The subjects responded to the questionnaires a median 117.5 days (25th to 75th percentile 105–135, range 41–200) after first symptom of COVID-19. In total, 117 (26%) of the respondents had contracted COVID-19 during travel abroad.

During the acute COVID-19 episode, the most common symptoms were fever (73%), loss/disturbance of taste (70%), dry cough (67%), headache (67%), and loss/disturbance of smell (66%). When aggregating the symptoms, 106 (23%) had 0–5 symptoms, 176 (38%) 6–10, and 176 (38%) 10–23.

### 3.2. Prevalence and Symptoms of Fatigue

The mean CFQ-11 bimodal score was 3.9 (SD 3.7) (*n* = 458); 246 (54%) scored 0–3 and 212 (46%) scored 4–11, corresponding to a case-definition of fatigue. Among women, 142 (55%) scored 4–11, compared with 70 (35%) among men (*p* < 0.001).

On the CFQ-11 with Likert scoring (*n* = 456), the mean total score (0–33 scale) was 15.1 (SD 5.0), the physical subscale (0–21 range) 10.1 (3.8), and mental subscale (0–12 range) 5.0 (1.8). The mean RAND-36 energy/fatigue scale score (0–100 range) was 56.8 (23.9) (*n* = 457).

The mean z scores of the CFQ-11 total was 0.70 (95% CI 0.58 to 0.82), CFQ-11 physical 0.66 (0.55 to 0.78), CFQ-11 mental 0.47 (0.35 to 0.59), and RAND-36 energy/fatigue −0.20 (−0.31 to −0.10), and all z scores were different from the norm population, i.e., different from 0 (*p* < 0.001) (Figure 2).

### 3.3. Determinants of Fatigue

In multivariable logistic regression analysis with fatigue (CFQ-11 bimodal score ≥ 4) as the dependent variable, male sex, being married/cohabiting, and longer time since symptom debut were associated with lower odds of fatigue at the time of the survey than the referents. In contrast, those with a higher symptom load during acute COVID-19 or confusion as a symptom had higher odds of persistent fatigue (Table 2).

In the multivariable regression analysis with CFQ-11 Likert scale score as the dependent variable, males had less fatigue symptoms, while previous depression, high symptom load during COVID-19, dyspnea during COVID-19, and confusion during COVID-19 had more fatigue symptoms at the time of the survey than those in the reference categories (Table 3).

Similarly, male sex and >127 days since symptom debut were associated with better health status on the energy/fatigue scale of the RAND-36, while higher BMI, previous depression, higher symptom load, dyspnea during COVID-19, and confusion during COVID-19 were indicators of worse health status than those in the reference categories. 

## 4. Discussion

In this study of non-hospitalized subjects, 46% of respondents reported fatigue about 4 months after symptom onset of COVID-19, which represents a substantially higher prevalence than in the norm population [20]. Fatigue was lower among males and higher with high symptom load and confusion during acute COVID-19 on three different fatigue scales. Moreover, previous depression, dyspnea during COVID 19, and higher BMI were associated with more fatigue on two of the scales. Finally, on two of the three scales, caseness of fatigue and symptoms of fatigue were lower among those with the longest follow up since symptom onset, than among referents.

The prevalence of fatigue in the present study was about twice as high as the 22% reported in a Norwegian general population, also using the CFQ-11 with the same cut-off to denote fatigue [20]. The continuous symptom scores in the present study were worse than in the same norm population, as illustrated by the z scores.

Fatigue is a common symptom in the general population, as well as in patients with acute or chronic diseases. It is poorly understood, and there is no consensus on its definition or causal mechanism [27]. There is wide variation in the prevalence of fatigue between studies depending on assessment methods, and thresholds, for fatigue [25,28,29]. There are numerous validated measures for assessment of fatigue [30], including the CFQ-11 used in the present study. However, some surveys after COVID-19 have used simple checklists, such as being asked to recount the presence or absence of symptoms, where fatigue was one of the symptoms, i.e., the presence of fatigue is indicated by the response to one yes/no item [13,31,32]. Therefore, assessments between studies may be difficult to compare.

There have been few studies on COVID-19 using standardized multi-item scales for assessing fatigue. We are only aware of one other community-based study, which used the fatigue impact scale and reported a prevalence of 24%, which is lower than in the present study, although it is not clear whether the prevalence numbers were from using this scale or a single item on fatigue [16]. The prevalence of fatigue in the present study was slightly lower than that reported in hospitalized and non-hospitalized individuals recovered from the acute phase of COVID-19 illness, where 52% reported persistent fatigue at a median of 10 weeks after initial symptoms of COVID-19 [17]. That study used the CFQ-11 and the same cut-off for defining fatigue as in the present study. Another study, using another multi-item scale, reported a prevalence of 53% about 4 weeks after hospitalization for COVID-19 [33], and in a third study 69% of respondents reported fatigue on the Nijmegen clinical screening instrument at 3 months after hospitalization for acute COVID-19 [34].

A few other studies have used single items to assess fatigue following COVID-19, generally reporting higher prevalence of fatigue than in our study [12,13,14,32]. However, differences in populations, methods, and timing relative to the acute phase complicates comparisons between studies. Fatigue was reported by 60–70% on average 48 days after hospital discharge [12], and by 53% of hospitalized patients on average 36 days after discharge (60 days after symptom start) (*n* = 143) [13]. In a large web-recruited population with hospitalized and non-hospitalized subjects after COVID-19 [32], 87% reported current fatigue on average 79 days after COVID-19, in a population mainly consisting of women. This finding of enduring fatigue, however, is not universal. For example, another study reported resolution of fatigue 3–4 weeks after discharge from hospital, although the number of subjects with fatigue was small [35]. Slow resolution of symptoms and persistent fatigue is not unique to COVID-19, as this has been reported following other infectious diseases, such as community-acquired pneumonia or SARS [36,37,38].

In the present study, persistent fatigue was associated with female sex, high symptom load, and confusion during acute COVID-19, as well as indications of an association with previous depression, dyspnea, and BMI. This supports and expands on previous findings that persistent fatigue following COVID-19 is associated with female sex and pre-existing depression/anxiety [17]. The latter study did not find an association of fatigue with time since symptom onset, which contrasts with the present study. This difference may be attributed to differences in populations, severity of disease, or length of the observation period.

Fatigue in the general population was more prevalent in women than in men in some studies [20,25], but not in others [28]. The association of fatigue with age has also been inconsistent [20,28]. Following SARS, those with fatigue symptoms after >6 months were more likely to have comorbid active psychiatric disorders [39]. Whether this applies to COVID-19 remains to be seen.

Some strengths of this study should be noted. The present study was large and was conducted in a geographical area comprising about 17% of the Norwegian population. As GPs and hospitals in the catchment area mainly use the three laboratories in the study, we anticipate that these labs will have analyzed >90% of the samples for SARS-CoV-2 in the two hospitals’ catchment populations. However, some patients may have been diagnosed in other laboratories during travel to other regions. As the study was population-based, it is more representative of the total population of non-hospitalized COVID-19 survivors than convenience samples, which increases the external validity of the findings. Other strengths are the comparison with general population norms, which facilitates interpretation of the scores, and that we used multivariable analysis, adjusting for several possible confounding variables.

Several limitations of the study should be noted. The overall response rate was about 50%, which we consider good for a population-based survey, and in line with a priori expectations. The response rate was higher than in some recent Norwegian large health surveys of 33–36%, using similar questionnaires in general populations [29,40]. Studies have shown a pattern of decreasing survey response rate over the past decades, although this depends on the length of the questionnaire, the topic, target population, and whether incentives are used [41,42,43]. Furthermore, web-surveys tend to have lower response rates than traditional mailed surveys [44].

The responses in our study were somewhat biased towards females and subjects >50 years of age. Moreover, the response rate was low in the three districts of Oslo with a large proportion of immigrants in the population. Some of this nonresponse may be related to limitations in knowledge of the Norwegian language, as the questionnaire was only available in Norwegian. This pattern of nonresponse is common in epidemiological surveys [29,41], and may cause nonresponse bias, which is difficult to control for and may limit generalizability. Responses to some of the questions, such as symptoms during COVID-19, may have been influenced by recall bias.

Surveys of the general population during the COVID-19 epidemic have shown a high prevalence of psychological morbidities and considerable distress during the period of lockdown, social distancing, and inability to travel, although with large variations between studies [45,46,47]. Recent studies have also shown an association between anxiety and somatic symptoms, including fatigue [48], and the World Health Organization has introduced the term “pandemic fatigue” to describe the feeling of distress in the population as a reaction to the prolonged period of uncertainty and crisis during the pandemic [49]. Therefore, in the present study, it is difficult to separate the general impact of the pandemic and the media exposure, from the impact of personal experience of the COVID-19 episode.

The high prevalence of fatigue following even mild cases of COVID-19 may have important consequences for individuals and society, although the observation time of 4 months is short. It is possible that rehabilitation or specific interventions may influence the development of these symptoms, although such interventions ought to be evaluated in randomized controlled trials. Those being most at risk for having persistent fatigue or other symptoms may possibly benefit from a tailored program of follow-up and support. If fatigue is accentuated by the threats of the pandemic, it is also possible that improvement or resolution of symptoms may occur spontaneously over time when the pandemic ends and social distancing eases.

## 5. Conclusions

This study has shown that persistent fatigue is common from 1.5–6 months after COVID-19 in a non-hospitalized population. The findings suggest that fatigue had started tapering off after about 4 months, which is promising. However, whether this represent the start of a long-term resolution of the symptoms should be investigated in longitudinal studies. Female sex, and high symptom load during acute COVID-19, which are non-modifiable risk factors, were markers of persistent fatigue symptoms.

## Figures and Tables

**Figure 1 ijerph-18-02030-f001:**
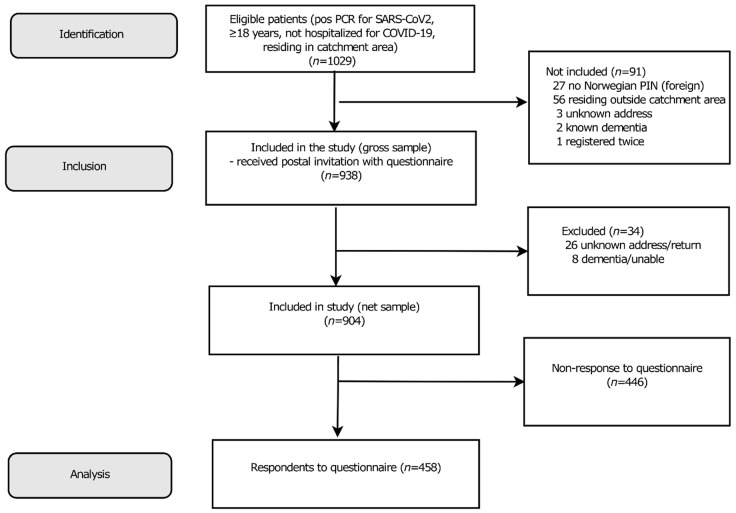
Flow chart showing the selection of the study population.

**Figure 2 ijerph-18-02030-f002:**
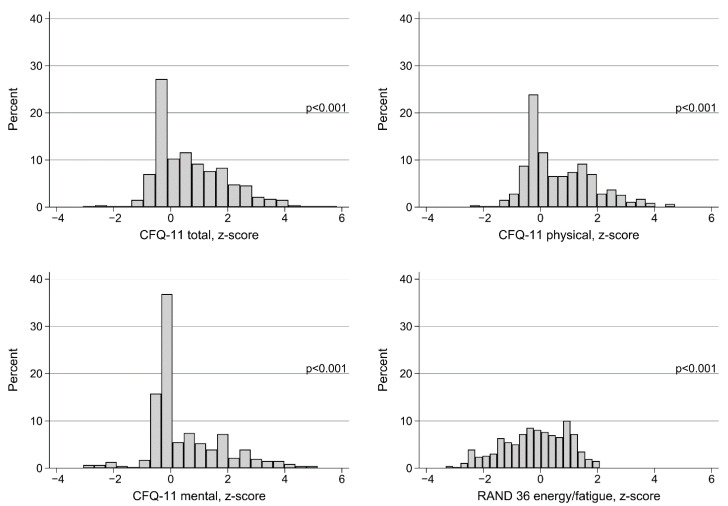
Fatigue and energy/fatigue scores at 1.5–6 months after start of COVID-19, z scores for comparison with age and sex-stratified normative data from Norwegian general populations. CFQ-11 = Chalder fatigue questionnaire. *p*-values are for comparison with z score = 0 (*t*-test).

**Table 1 ijerph-18-02030-t001:** Respondent characteristics (*n* = 458). Number (%), unless specified otherwise.

Age, Mean (Range)	49.6 (17.7 to 87.9)
Sex, female	256 (56)
Marital status	
Single/divorced/separated/widowed	121 (26)
Married/cohabiting	336 (74)
Born in Norway (*n* = 457)	382 (84)
Norwegian mother tongue (*n* = 455)	382 (84)
Both parents born in Norway (*n* = 434)	357 (78)
Highest attainted education	
Primary school	41 (9)
Secondary school	174 (38)
University	243 (53)
Smoking status (*n* = 453)	
Never smoker	298 (66)
Former/current smoker	155 (34)
Body mass index (kg/m2) (*n* = 450)	26.8 (5.2)
Influenza vaccination 2nd half of 2019 (*n* = 456)	141(31)
No. of 21 comorbidities, categorized	
0	234 (51)
1	129 (28)
≥2	95 (21)
Diabetes	16 (3)
Pulmonary disease (asthma, COPD, other)	60 (13)
Depression	30 (7)
Cardiovascular (heart, hypertension, vascular)	101 (22)
Test lab	
Akershus University Hospital	233 (51)
Fürst Laboratory	61 (13)
Østfold Hospital	164 (36)

COPD = chronic obstructive pulmonary disease.

**Table 2 ijerph-18-02030-t002:** Determinants of fatigue (defined as CFQ-11 bimodal score, >3 vs. 0–3), multivariable logistic regression analysis (*n* = 440).

Variable	*n*	Odds Ratio	95% Confidence Interval	*p*
Age, per 10 years	440	1.02	(0.86 to 1.22)	0.81
Sex				
Female *	245	1		
Male	195	0.49	(0.31 to 0.76)	0.002
Marital status				
Single/separated/divorced/widowed *	112	1		
Married/cohabiting	328	0.56	(0.34 to 0.92)	0.022
Highest attained educational level				
Primary school (≤11years)	37	1		
Secondary school (12–13 years)	165	1.22	(0.54 to 2.74)	0.64
University level	238	1.17	(0.53 to 2.61)	0.70
No. of comorbidities (out of 21)				
0 *	223	1		
1	123	1.62	(0.94 to 2.77)	0.080
≥2	94	1.52	(0.77 to 3.03)	0.23
Previous depression				
No *	412	1		
Yes	28	1.10	(0.43 to 2.82)	0.84
No. of COVID-19 symptoms				
0–5 *	101	1		
6–9	168	1.44	(0.79 to 2.64)	0.24
10–23	171	3.66	(1.88 to 7.11)	<0.001
Dyspnea during COVID-19				
No *	190	1		
Yes	250	1.56	(0.97 to 2.53)	0.069
Confusion during COVID-19				
No *	381	1		
Yes	59	2.25	(1.12 to 4.51)	0.022
Body mass index, kg/m^2^	440	1.03	(0.99 to 1.08)	0.13
Smoking status				
Never smoker *	291	1		
Former/current smoker	149	1.34	(0.85 to 2.13)	0.21
Time since symptom onset, days				
41–110 *	144	1		
111–127	152	0.80	(0.47 to 1.36)	0.41
128–200	144	0.55	(0.32 to 0.96)	0.034

* Baseline category. CFQ-11 = Chalder fatigue questionnaire.

**Table 3 ijerph-18-02030-t003:** Determinants of fatigue scores, (1) CFQ-11 total score (0–33 scale) ^a^ and (2) RAND-36 energy/fatigue scale score (0–100 scale) ^b^, multiple linear regression analysis.

Variable	CFQ-11 Total Score (*n* = 438)			RAND-36 Energy/Fatigue (*n* = 440)		
	Coef.	95% Confidence Interval	*p*	Coef.	95% Confidence Interval	*p*
Age, per 10 years	0.1	(−0.24 to 0.44)	0.56	1.51	(−0.05 to 3.07)	0.057
Sex						
Female *	0			0		
Male	−1.78	(−2.66 to −0.90)	<0.001	9.63	(5.58 to 13.69)	<0.001
Marital status						
Single/separated/divorced/widowed *	0			0		
Married/cohabiting	−0.84	(−1.81 to 0.12)	0.086	3.53	(−0.93 to 7.99)	0.12
Highest attained educational level						
Primary school (≤11 years)	0			0		
Secondary school (12–13 years)	0.03	(−1.55 to 1.62)	0.97	3.98	(−3.37 to 11.33)	0.29
University level	−0.03	(−1.58 to 1.52)	0.97	4.42	(−2.77 to 11.62)	0.23
No. of comorbidities (out of 21)						
0 *	0			0		
1	0.48	(−0.57 to 1.53)	0.37	−1.35	(−6.24 to 3.54)	0.59
≥2	0.22	(−1.13 to 1.57)	0.75	−6.11	(−12.35 to 0.14)	0.055
Previous depression						
No *	0			0		
Yes	2.38	(0.57 to 4.18)	0.010	−12.05	(−20.43 to −3.68)	0.005
No. of COVID-19 symptoms						
0–5 *	0			0		
6–9	0.70	(−0.44 to 1.85)	0.23	−8.28	(−13.58 to −2.98)	0.002
10–23	2.68	(1.38 to 3.99)	<0.001	−15.59	(−21.64 to −9.55)	<0.001
Dyspnea during COVID-19						
No *	0			0		
Yes	1.24	(0.29 to 2.19)	0.010	−6.12	(−10.53 to −1.72)	0.007
Confusion during COVID-19						
No *	0			0		
Yes	2.65	(1.34 to 3.97)	<0.001	−7.35	(−13.44 to −1.26)	0.018
Body mass index, kg/m^2^	0.04	(−0.04 to 0.12)	0.33	−0.50	(−0.88 to −0.12)	0.010
Smoking status						
Never smoker *	0			0		
Former/current smoker	0.66	(−0.25 to 1.56)	0.15	−3.91	(−8.10 to 0.28)	0.068
Time since symptom onset, days						
41–110 *	0			0		
111–127	−0.56	(−1.60 to 0.47)	0.28	1.38	(−3.40 to 6.17)	0.57
128–200	−0.41	(−1.47 to 0.64)	0.44	6.09	(1.20 to 10.99)	0.015

* Baseline category. CFQ-11 = Chalder fatigue questionnaire; Coef. = Unstandardized beta coefficient; ^a^ Higher values mean more symptoms, ^b^ Higher values denote better health status.

## Data Availability

Anonymized data supporting the study findings are available from the corresponding author on reasonable request.

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
