# Peer review of "Prevalence and Determinants of Fatigue after COVID-19 in Non-Hospitalized Subjects: A Population-Based Study"

_ijerph, 2021, doi:10.3390/ijerph18042030_

Round 1

Reviewer 1 Report

I have several concerns regarding this study.

First of all, I am not entirely sure of its usefulness in terms of it's findings. The main conclusion is that females who are single/divorced/widowed, who had a short time symptom debut, high symptom load and confusion were most likely to have fatigue. The latter three are obvious - the former two have been discussed elsewhere as pointed out in their discussion. There is no discussion on how are policymakers to act on this knowledge and improve population health. The usefulness of these findings is very limited at this stage of the outbreak where movement is heading towards mitigation.

Are females who are single/divorced/widowed generally more fatigued day to dat in their population sample? How can this effect be attributed specifically to COVID? Are there interactions at play here in terms of age versus education, or age and symptom load?

What is most concerning is the shortness of the limitations section which does not examine in sufficient detail the many shortcomings of this study in its assumptions regarding how representative the questionnaire is - indeed the research on perceptions of COVID have shown great variability and general increase in stress population wide. General stress from lockdowns, social distancing and the inability is travel will be difficult to discern from fatigue.

Self reported comorbidities is a critical issue - it is well established that comorbidities is a key indicator of severity of symptoms. Diabetes or pre-diabetics are most likely heavily underrepresented.

Recall bias is likely to be a substantial problem in their symptomatic date and other symptoms reported, which is not mentioned.

Author Response

Reviewer1

I have several concerns regarding this study.

First of all, I am not entirely sure of its usefulness in terms of it's findings. The main conclusion is that females who are single/divorced/widowed, who had a short time symptom debut, high symptom load and confusion were most likely to have fatigue. The latter three are obvious - the former two have been discussed elsewhere as pointed out in their discussion. There is no discussion on how are policymakers to act on this knowledge and improve population health. The usefulness of these findings is very limited at this stage of the outbreak where movement is heading towards mitigation.

This study has tried to document the fatigue many patients perceive after COVID-19, including presenting the incidence in a defined population. We do not think this is that obvious, as there is not much documentation on this in mild COVID-19 or non-hospitalized subjects using robust methods.

  • We have stated the objective of the study more clearly in the last paragraph of the Introduction (lines 57-60).
  • We have added a comment on possible implications of this knowledge towards the end of the discussion (lines 295-303):

Are females who are single/divorced/widowed generally more fatigued day to dat in their population sample? How can this effect be attributed specifically to COVID? Are there interactions at play here in terms of age versus education, or age and symptom load?

There was no item on day to day variation in symptoms. The CFQ-11 asks whether they have more symptoms than usual. Each of the 11 items are answered on a 4-point scale ranging from the asymptomatic to maximum symptomology, such as ‘Better than usual’, ‘No worse than usual’, ‘Worse than usual’ and ‘Much worse than usual’.

Because of the cross-sectional design, we cannot be sure that the perceived fatigue can be attributed to COVID-19, although we think this is likely.

  • We have expanded the section on this in the limitations section in the Discussion (lines 283-294)
  • There was no statistical interaction in any of the models of fatigue or fatigue scores in terms of age*education or age *symptom load.

What is most concerning is the shortness of the limitations section which does not examine in sufficient detail the many shortcomings of this study in its assumptions regarding how representative the questionnaire is - indeed the research on perceptions of COVID have shown great variability and general increase in stress population wide. General stress from lockdowns, social distancing and the inability is travel will be difficult to discern from fatigue.

We agree and have expanded the section on the limitations in the Discussion (lines 269-294).

Self reported comorbidities is a critical issue - it is well established that comorbidities is a key indicator of severity of symptoms. Diabetes or pre-diabetics are most likely heavily underrepresented.

We agree that comorbidities are important in understanding symptoms and effects of disease in general. Therefore, we have adjusted for this in the analyses, using a standardized measure of comorbidity.

In Norway, the overall prevalence if type diabetes in Norway is estimated to 4.7%, of which 90% have diabetes type 2 (https://www.fhi.no/en/op/hin/health-disease/diabetes-in-norway---public-health-/). The study population of non-hospitalized subjects with COVID-19 would be expected to have a lower prevalence of diabetes than the general population because of a larger proportion of young subjects, and because elderly and patients with diabetes and COVID-19 would be more likely to be hospitalized. A recent report based on Norwegian registry data reported 9.4% of diabetes among 1025 patients hospitalized for COVID-19 (Nystad et al, 2020). Therefore, a prevalence of diabetes of 3% in the sample seems reasonable.

Nystad W, Hjellvik V, Larsen IK, Ariansen I, Helland E, Johansen KI, et al. Underlying conditions in adults with COVID-19. Tidsskr Nor Laegeforen. 2020;140(13).

Recall bias is likely to be a substantial problem in their symptomatic date and other symptoms reported, which is not mentioned.

We agree that recall bias represents a challenge for symptoms at the time of COVID-19. The test date, however, is from the labs and is consistent with the dates of symptom start reported by the patients. We have expanded the text on limitations in the Discussion (line 283-284).

Reviewer 2 Report

With interest I read the paper by Stavem et al. addressing post-COVID-fatigue.

This is a well performed study with a clear introduction, applying the appropriate methodology to cover a clinical and epidemiological serious problem and clearly presented results.

However, I think that the authors miss the opportunity to discuss their findings in a broader, more inspiring and maybe even speculative context (e.g. cognitive, physical and emotional impact of fatigue), which will increase the interest to the readers. If some of the figures and tables are presented as online-supplementary material, it should not be a problem to add up to 1 extra page of text to the discussion:

  • They compare their found prevalence with a normal population from a paper from 1998  and miss to address any psychological / economical factors (lockdown!), which might be relevant and contributing to this problem as well.
  • "Pandemic fatigue" as problem could also be addressed in the discussion: (PMID: 33362178). I further recommend to add PMID 33352638, PMID: 33220049, PMID: 33413026 and PMID: 33144403 as references to the discussion and to mirror their findings with the findings by the authors.
  • This should also enable the authors to make some final recommendations for a future research agenda.

Finally, the reference list requires a careful revision, because there are several inconsistencies in the way of presentation, especially of the page-numbers.

Author Response

Reviewer 2

With interest I read the paper by Stavem et al. addressing post-COVID-fatigue.

This is a well performed study with a clear introduction, applying the appropriate methodology to cover a clinical and epidemiological serious problem and clearly presented results.

However, I think that the authors miss the opportunity to discuss their findings in a broader, more inspiring and maybe even speculative context (e.g. cognitive, physical and emotional impact of fatigue), which will increase the interest to the readers. If some of the figures and tables are presented as online-supplementary material, it should not be a problem to add up to 1 extra page of text to the discussion:

  • They compare their found prevalence with a normal population from a paper from 1998  and miss to address any psychological / economical factors (lockdown!), which might be relevant and contributing to this problem as well.
  • "Pandemic fatigue" as problem could also be addressed in the discussion: (PMID: 33362178). I further recommend to add PMID 33352638, PMID: 33220049, PMID: 33413026 and PMID: 33144403 as references to the discussion and to mirror their findings with the findings by the authors.
  • This should also enable the authors to make some final recommendations for a future research agenda.

Thanks for constructive comments to the paper. We have expanded the discussion to address the above points, including the suggested references (lines 269-294).  

We had already referred to one of the papers (PMID: 33413026), which has the population that probably is most similar to our cohort. We are also aware of van den Borst et al (PMID: 3314403) which concerns hospitalized patients.

Finally, the reference list requires a careful revision, because there are several inconsistencies in the way of presentation, especially of the page-numbers.

Thanks. The reference list has been reviewed and updated by using the most recent endnote style for MDPI.

Reviewer 3 Report

The authors describe the Prevalence and determinants of fatigue after COVID-19 in non-hospitalized subjects. I have few concerns regarding the study which need to be addressed:

  1. Overall, the manuscript need extensive English editing. There are too many spelling and grammar errors.
  2. The authors focused on non-hospitalized patients ≥18 years, what about other age below 18?
  3. Line 32-35: need references.
  4. Line 37: in case the authors want to compare between SARS and COVID-19 in respect to clinical symptoms, I would rather recommend including MERS as well.
  5. Line 48: the authors mentioned that ‘the above studies used single items for assessing fatigue’ what are these items? Please describe in details to avoid the reader’s confusion.
  6. Line 65: Until 1 June 20202: again mistake, please revise all the manuscript.
  7. Line 71: The subjects, did the authors mean patients? Please clarify throughout the manuscript.
  8. Line 72: consent form and questionnaire, can the authors provide a draft for these forms as supplementary figures?
  9. The number of patients who did not respond to the questionnaire are too high (n=446), what the authors think about reasons for not responding or what were the obligations?
  10. Table 1 is confused and the subheadings are not clear.
  11. Table 2 should has more footnotes to describe the abbreviations within the table.
  12. Table 3: the authors mentioned within the methodology that the surveillance include only ages more than 18 while within the table they include primary schools, can you please explain?
  13. Discussion is so poor and lack showing the study novelty. Need editing and try to avoid redundancy between results and discussion sections. Otherwise, try to combine both sections.

Author Response

Reviewer 3

The authors describe the Prevalence and determinants of fatigue after COVID-19 in non-hospitalized subjects. I have few concerns regarding the study which need to be addressed:

  1. Overall, the manuscript need extensive English editing. There are too many spelling and grammar errors.
  2. The authors focused on non-hospitalized patients ≥18 years, what about other age below 18?
  3. Line 32-35: need references.
  4. Line 37: in case the authors want to compare between SARS and COVID-19 in respect to clinical symptoms, I would rather recommend including MERS as well.
  5. Line 48: the authors mentioned that ‘the above studies used single items for assessing fatigue’ what are these items? Please describe in details to avoid the reader’s confusion.
  6. Line 65: Until 1 June 20202: again mistake, please revise all the manuscript.
  7. Line 71: The subjects, did the authors mean patients? Please clarify throughout the manuscript.
  8. Line 72: consent form and questionnaire, can the authors provide a draft for these forms as supplementary figures?
  9. The number of patients who did not respond to the questionnaire are too high (n=446), what the authors think about reasons for not responding or what were the obligations?
  10. Table 1 is confused and the subheadings are not clear.
  11. Table 2 should has more footnotes to describe the abbreviations within the table.
  12. Table 3: the authors mentioned within the methodology that the surveillance include only ages more than 18 while within the table they include primary schools, can you please explain?
  13. Discussion is so poor and lack showing the study novelty. Need editing and try to avoid redundancy between results and discussion sections. Otherwise, try to combine both sections.

Thanks for specific and useful comments.

  1. Spelling errors and apparent grammatical errors have been corrected in the revised manuscript
  2. This study only included adults, defined as those above 18 years of age. Therefore, we have no data on those below 18 years of age.
  3. References have been added
  4. Agree, we have included MERS as well and added a reference.
  5. “Single item” here refers to using a single question to report fatigue, such as “Do you feel fatigued” Yes/No or a checklist for checking if you have the symptom now. We have added this with more details to the discussion, including references) (lines 216-220)
  6. This error has been corrected (line 67)
  7. The participants in the survey had COVID-19 some months before, and would be considered patients at that time. However, they were recruited based on a positive PCR-test, and would be expected to recover at the time of the survey. Moreover, they had not been in contact with us as patients. Therefore, we think it is more appropriate to refer to the respondents in the survey as subjects, participants, or respondents in the survey, rather than patients, However, when we refer to the acute COVID-19 episode they were patients. We think naming them patients could mean stigmatizing the respondents, and some of them may be sensitive to this.
  8. We have added a reproduction of the 3-page electronic consent form as a supplementary figure, which can be added as a supplement at the editor’s discretion. The questionnaire contained 15 pages in Norwegian (alternatively an electronic version), and we think these have limited interest. The fatigue questionnaire used is available elsewhere;

https://helse-bergen.no/seksjon/sovno/Documents/Fatigue_Questionnaire_norsk%202.pdf

  1. The response rate to the questionnaire was 49%. This is at the level we a priori thought was realistic, given that we only were allowed to send one reminder and no telephone follow-up. In the study protocol in May 2020, we stated “It is difficult to estimate the response rate in such a survey. In this population, there are many subjects with insufficient knowledge of the Norwegian language. We think that a response rate of about 50% (range 35–60%) could be achievable after one reminder.”

Response rates today are generally lower than a few years back in such surveys. Because of the focus on COVID-19 in the media, we expected a higher response rate than that. We have limited information on non-respondents. We know they were younger and comprised more males. In addition, the response rate was very low in areas with 40-60% immigrants, where also the population has less education and lower socioeconomic status. In this area, probably Norwegian language skills and general skepticism to authorities may play a role.

We have expanded the discussion of this point (lines 269-276).

  1. Agree, however, we have used the publisher’s template in the formatting (which centers the labels). Added explanation to abbreviation used.
  2. We added explanation of CFQ-11 as a footnote and spelled confidence interval out in the heading.
  3. This clearly is a misunderstanding. This is about the level of education of the respondents, where the lowest level is having completed the Primary (compulsory) school level. This is listed under the subheading “Highest attained educational level” in the table.
  4. We have expanded parts of the discussion to emphasize what is new and the implications. We have summarized the main findings of the study in the first part of the discussion, however, have tried to avoid unnecessary repetition of the results elsewhere in the discussion, as suggested.

Round 2

Reviewer 1 Report

Limitations have been expanded appropriately to accommodate for the methodological concerns and errors highlighted have been rectified. The paper still unfortunately remains of little interest to the journal's audience or policymakers as a whole.

Reviewer 3 Report

Dear authors, thanks for responding to my comments.